# Effect of Dispersion Medium on Pharmacokinetic Profile of Rotigotine Crystalline Suspension following Subcutaneous Injection

**DOI:** 10.3390/pharmaceutics14122630

**Published:** 2022-11-28

**Authors:** Min Seop Kim, Myoung Jin Ho, Min Yeong Joung, Yong Seok Choi, Myung Joo Kang

**Affiliations:** College of Pharmacy, Dankook University, 119 Dandae-ro, Dongnam-gu, Cheonan 330-714, Republic of Korea

**Keywords:** Rotigotine, parenteral long-acting delivery, crystalline suspension, dispersion medium, pharmacokinetics, local distribution, local inflammation

## Abstract

Rotigotine (RTG) is prescribed as a once-daily transdermal patch for managing early Parkinson’s disease (PD), which presents issues such as skin irritation and poor patient adherence. Therefore, the aims of the present study were to formulate aqueous and oily vehicle-based RTG crystalline suspensions for prolonged delivery and to compare their pharmacokinetic profiles and the local behaviors of RTG crystals. RTG-loaded aqueous (AS) and oil suspensions (OS) were fabricated using bead-milling technology (100 mg/mL as RTG), employing carboxymethyl cellulose and sesame oil as suspending agent and oily vehicle, respectively. RTG AS and OS exhibited comparable physical properties in terms of particle size (about 800–900 nm), crystallinity, and dissolution profile, despite higher drug solubility in OS than AS (19.6 and 0.07 mg/mL, respectively). However, AS and OS exhibited markedly distinctive local distribution and inflammatory responses at the injection site, which further promoted different pharmacokinetic patterns following subcutaneous injection in rats. With OS, no drug aggregates were observed with prolonged persistence of the Sudan III-stained oily vehicle at the injection site. In contrast, with AS injection, drug clusters > 7 mm were formed, followed by an enclosure with macrophages and a fibroblastic band. Accordingly, AS exhibited a protracted pharmacokinetic profile over 3 weeks, with prolonged elimination half-life. The local inflammatory response caused by AS injection was almost alleviated after 3 weeks post-dosing. Based on these findings, we conclude that RTG AS system can be a platform to design sophisticated long-acting delivery systems with extended dosing intervals to manage PD.

## 1. Introduction

Rotigotine ([–]2-(N-propyl-N-2-thienylethylamino)5-hydroxytetralin, RTG), a non-ergot dopamine agonist, has been prescribed to manage early Parkinson’s disease (PD) as monotherapy or an adjunct to levodopa [1,2]. Because of its extensive first-pass metabolism in the intestine and liver [3], the lipophilic dopamine agonist was formulated as a once-daily transdermal patch (Neupro^®^, Schwarz Pharma, Brussels, Belgium). The transdermal system provides steady absorption of the drug molecule over 24 h, reducing systemic adverse effects, such as motor fluctuations, and dyskinesia [2,4]. However, repeated application of adhesive patches to the skin causes adverse reactions at the application site in patients with PD. In clinical trials, approximately half of RTG patients had application site reactions, including erythema, swelling, and pruritus [4]. Moreover, considering the poor drug compliance by patients with PD, an alternative dosage form offering appropriate therapeutic levels over several weeks following a single administration can provide the benefit of reduced dosing frequency [5,6].

The long-acting injectable (LAI) drug delivery system based on a crystalline suspension is in the spotlight as an effective tool to afford an extended pharmacokinetic profile following a single administration, thereby extending the dosing intervals for patients with chronic diseases [7,8,9]. A crystalline suspension is defined as a carrier-free colloidal or coarse dispersion of water-insoluble drug particles with a minimal quantity of steric stabilizers in the dispersion vehicles. This parenteral delivery system provides high drug loading capacity, effective modulation of pharmacokinetic profile, and ease of scale-up [9,10]. Following intramuscular (IM) or subcutaneous (SC) injections, insoluble drug particles are steadily dissolved at the local tissue and are absorbed into the bloodstream, resulting in a continuous pharmacokinetic profile over several weeks. Drug crystals can also be absorbed by inflammatory cells, including macrophages, enter into lymphatic capillaries, and can subsequently be transferred from the lymphatic system into the bloodstream [11,12]. The pharmacokinetic profile and local tolerability of drug crystals at the injection site depend on several formulation factors, such as drug concentration, particle size, surface properties, and particle elasticity, as well as the nature of the dispersion medium [13,14,15,16,17]. The spread and distribution of drug particles along subcutaneous fibrils vary depending on the viscosity and affinity of the delivery vehicle with interstitial fluid and tissues, which influences the interfacial area between the drug particles and aqueous media. Moreover, aqueous and oily vehicles have different disappearance rates at injection sites; oily vehicles typically have retarded disappearance rates at injection sites [14,18]. Thus, the type of dispersion medium can be a key factor in varying the spreading, aggregation, dissolution, and biological response against drug particles at the injection site. However, to the best of our knowledge, studies have yet to evaluate the effect of parenteral delivery vehicles on the behavior of drug particles at the injection site and the pharmacokinetic profile.

The goals of the present study were to formulate an aqueous suspension (AS) and oil suspension (OS) of RTG and to comparatively evaluate the in vivo pharmacokinetic profile, along with the local behavior of drug particles at the injection site. RTG-loaded AS and OS were fabricated using a laboratory-scale bead-milling method, and their physicochemical properties such as morphology, particle size, viscosity, and in vitro dissolution profile were characterized. The pharmacokinetic profile of the RTG-loaded injectable suspension systems, depending on the delivery vehicle, was assessed in rats using LC-MS/MS analysis. Moreover, to understand the pharmacokinetic behavior of RTG particles depending on the delivery vehicle, the distribution and inflammatory response of drug particles at the injection site were macroscopically and histopathologically assessed in rats following SC injection.

## 2. Materials and Methods

### 2.1. Materials

The RTG-free base was obtained from Ferrer Pharmaceutical Co., Ltd. (Barcelona, Spain). Sodium carboxymethyl cellulose (Na. CMC), polyethylene glycol 4000 (PEG 4000), soybean oil, corn oil, peanut oil, potassium phosphate dibasic, and triethylamine were purchased from Sigma-Aldrich (St. Louis, MO, USA). Polysorbate 80, castor oil, and cottonseed oil were provided by Croda Korea (Seoul, Korea). Polyoxyethylene (160), polyoxypropylene (30) glycol (Poloxamer 188), polyoxyl 15 hydroxystearate (Kolliphor HS 15), polyoxyl 40 hydrogenated castor oil (Kolliphor RH 40), polyoxyl 35 castor oil (Kolliphor EL), and polyvinylpyrrolidone K17 (PVP K17) were provided by BASF (Ludwigshafen, Germany). Medium-chain triglycerides (Miglyol 810N) and tricaprylin (Captex 8000) were obtained from IOI Oleo GmbH (Witten, Germany) and Abitec Corporation (Columbus, OH, USA), respectively. Sodium metabisulfite was purchased from Daejung Chemicals & Metals (Siheung-si, Korea). HPLC-grade methanol was purchased from J.T. Baker (Phillipsburg, NJ, USA). All other reagents or chemicals were of analytical grade and were employed without further purification.

### 2.2. Preparation of RTG Crystalline Suspensions Using Bead-Milling Technology

RTG-loaded AS and OS were prepared by crushing the raw material into fine drug crystals in an aqueous or oily media using a laboratory-scale bead-milling apparatus [19,20,21]. To formulate RTG-loaded AS, an aqueous vehicle was first prepared by dissolving 50–300 mg of suspending agents (Table 1, 0.05–0.3% *w*/*v*), 900 mg (0.9% *w*/*v*) of sodium chloride as an isotonic agent, and 50 mg (0.05% *w*/*v*) of sodium metabisulfite as an antioxidant in 100 mL of phosphate-buffered solution (15 mM, pH 7.0). Then, 1 mL of the aqueous vehicle, 100 mg of drug powder, and 1 g of 0.3 mm zirconia beads were put into a 2 mL Eppendorf tube. After pre-wetting for 1 min using a vortex shaker at 25 °C, the mixture was pulverized using the ZentriMix 380R (Andreas Hettich GmbH und Co KG, Tuttlingen, Germany) at different velocities (500, 1000, and 1500 rpm) for 1.5 h. During the milling process, the temperature of the cooling system was set to −10 °C to maintain a sample temperature below 30 °C, preventing thermal degradation of RTG during fabrication. The preparation of OS was analogous to that of AS; 100 mg of RTG power, and 1 g of 0.3 mm beads were contained to the different oily vehicles (1 mL, Table 2) and bead-milled at 1000 or 2000 rpm for 0.5 or 1.5 h. The prepared AS and OS were separated from the zirconia beads and stored at 25 °C in light-protective scintillation vials.

### 2.3. Characterization of RTG Crystalline Suspensions

#### 2.3.1. Morphological Observations

The morphological features of the drug crystals dispersed in the vehicles were evaluated using cryo-field emission transmission electron microscopy (Cryo-TEM, Glacios, Thermo Fisher, Waltham, MA, USA). Approximately 2 µL of RTG-loaded AS and OS was added dropwise to the Cu grid, and the samples were treated using Vitrobot (Mark IV, FEI, Hillsboro, OR, USA). The samples were then frozen using liquid nitrogen and observed at an acceleration voltage of 200 kV.

#### 2.3.2. X-ray diffractometry (XRD)

The XRD patterns of RTG crystals dispersed in aqueous and oily vehicles were chronicled using an X-ray diffractometer (Ultima IV, Rigaku Corporation, The Woodlands, TX, USA). As a control, raw materials and aqueous and oily vehicles were also determined. To harvest the RTG crystals dispersed in the vehicles, each drug suspension was centrifuged at 13,000 rpm for 10 min. The collected RTG nanocrystals were transferred to a stainless-steel dish and desiccated at room temperature for 24 h. Each sample was placed on a flat aluminum sample holder, and the diffraction pattern was scanned using CuKα radiation with λ = 1.54 Å (40 kV and 35 mA) in the range of 5° to 40° with a step size of 0.02°.

#### 2.3.3. Drug Content Analysis

The drug content in the AS and OS systems was determined by HPLC analysis [22]. RTG-loaded AS and OS (1 mL) were dissolved in 9 mL of methanol or hexane:2-propanol mixture (9:1 *v*/*v*), respectively, and 100-fold diluted with the mobile phase for HPLC analysis. To determine the amount of RTG dissolved in the aqueous or oily vehicle, 1 mL of RTG-loaded suspensions were centrifuged at 13,000 rpm for 10 min, to settle drug crystals, and the supernatant was collected and pretreated in the same manner. The amount of RTG suspended in the formulation was calculated by subtracting the content of RTG dissolved in the vehicle from the total content of RTG in the crystalline suspensions.

The concentration of RTG in each sample was determined using HPLC (Shimadzu Prominence). The HPLC system consisted of a pump (LC-20AD), a UV–VIS (ultraviolet–visible) detector (SPD-20A), an autosampler (CBM-20A), and a column oven (CTO-20AC). The mobile phase consisted of phosphate buffer (10 mM, pH 5.5, 0.2% *v*/*v* triethylamine) and methanol at a volume ratio of 3:7. The mobile phase was passed through a C18 column (4.6 mm × 150 mm, 5 µm, Phenomenex, Torrance, CA, USA) maintained at 40 °C at a flow rate of 1.0 mL/min. The wavelength of the UV detector eluent was set to 225 nm with a retention time of RTG at 6.57 min. The calibration curve drawn between RTG concentration and area under the peak was lined (y = 53795x + 926.19, *r*^2^ = 1) in the range of 0.5–200 μg/mL. The limit of detection (LOD) and limit of quantitation (LOQ) of HPLC analysis method were 0.034 μg/mL and 0.104 μg/mL, respectively.

#### 2.3.4. Determination of Crystal Size of RTG Crystalline Suspensions

The particle size and zeta potential of the RTG AS were analyzed using a Zetasizer Nano^®^ Instrument (Malvern Instruments, Malvern, UK) [23,24]. RTG-loaded AS was 80-fold diluted with distilled water and then loaded onto capillary cells. The size distribution of NS was analyzed at a wavelength of 633 nm, with a scattering angle of 90°.

For the RTG-OS, the size of the suspended RTG crystals was determined using an optical microscope (Eclipse 80i, NIKON, Tokyo, Japan), because of difficulty analyzing the hydrodynamic diameter with Zetasizer Nano^®^. Approximately 10 µL of the drug suspension was added dropwise to the slide glass and observed at 400× magnification. From the images, 200 drug particles were randomly selected and the length of the major axis was measured using a micrometer. The mean and standard deviation were calculated from three measurements per formulation [25].

#### 2.3.5. Viscosity of RTG Crystalline Suspensions

The apparent viscosity of the RTG-loaded crystalline suspensions was determined using a rotational rheometer (ARES-G2, TA Instrument Ltd., New Castle, DE, USA) furnished with a parallel plate (40 mm diameter plates) [26]. Approximately 2 g of injectable suspension was added to the lower plate, and the upper and lower plates were spaced 1 mm apart. The samples were then subjected to the desired shear rate (10–1000 s^−1^). The temperature was set at 25 °C with a temperature accuracy of ±0.1 °C. The tolerance for each measurement was set at 5%.

### 2.4. In Vitro Dissolution Profile of RTG Crystalline Suspensions

The in vitro dissolution profiles of RTG from AS and OS were compared using the dialysis membrane method [27,28]. To provide sink conditions and miscibility of the oily vehicle with the dissolution media, a mixture of buffered saline (10 mM, pH 7.4) and 2-propanol at a ratio of 60:40 (*v*:*v*) was employed as the dissolution medium. Approximately 200 µL of drug suspension containing 20 mg of RTG was loaded into a Float-A-Lyzer^®^ (50 kDa MWCO, Spectrum Lab, CA, USA). Drug-loaded Float-A-Lyzer^®^ was then placed in a beaker containing 200 mL of the pre-warmed dissolution media (37 ± 0.5 °C) and stirred at 250 rpm. At predetermined times, the dissolution medium (about 1 mL) was withdrawn and centrifuged at 13,000 rpm for 5 min. Prewarmed fresh dissolution media (37 °C) was replaced in a glass container after sampling. Subsequently, 500 µL of the supernatant was 2-times diluted with the mobile phase and the drug content in the sample was analyzed using the HPLC method described in Section 2.3.3. The similarity of the dissolution profiles of AS and OS was further compared by fitting the following equation [29]:$$f_{2}=50\cdot \log\left\{{\left[1+\frac{1}{n}\overset{n}{\underset{t=1}{\sum }}{\left(AS_{t}-OS_{t}\right)}^{2}\right]}^{-0.5}\times 100\right\}$$
where $f_{2}$ is the similarity factor, n is the number of time points, and $AS_{t}$ and $OS_{t}$ are the dissolution values of AS and OS at time t, respectively. The dissolution profiles between two formulas are considered to be analogous when the $f_{2}$ value ranges between 50 and 100 [29].

### 2.5. In Vivo Macroscopic and Histological Observations of Injected Sites following SC Injection

To visualize the injection site, RTG OS and AS were stained with Sudan III (0.1% *w*/*v*) and trypan blue (0.02% *w*/*v*), respectively. The stained RTG OS and AS formulas were administered subcutaneously to the rat’s back scruff tissues at a dose of 30 mg/kg using a 26-gauge BD syringe. At predetermined times (0.5 h and 4 days post-dosing), subcutaneous tissues were surgically isolated and were fixed in neutral-buffered formalin (10% *v*/*v*) for 48 h at 4 °C. Subsequently, fixed tissues were washed with 1X phosphate-buffered saline and stored in 30% *w*/*v* sucrose solution in 1X phosphate-buffered saline at 4 °C. Dehydrated tissues were trimmed to load into the cryomold with optimal cutting temperature (OCT) compounds. Frozen tissues at −28 °C were then cross-sectioned using a cryostat cryocut microtome (Model CM3050S, Leica, Wetzlar, Germany) at a thickness of 20 µm [30,31,32].

In vivo degree of inflammation in subcutaneous tissue following a single injection of RTG suspensions was further evaluated in six-week-old male normal Sprague-Dawley rats (200 ± 20 g). This study was approved by the Institutional Animal Care and Use Committee (IACUC) of Dankook University (approval number: DKU-19-033, date of approval: 8 October 2019). After at least 3 days acclimatization time, RTG suspensions (RTG AS and OS) were injected into the subcutaneous regions of the back scruff at a dose of 30 mg/kg, which was identical to the drug dose for next pharmacokinetic evaluation. At predetermined times (2, 7, and 21 days after administration), the rats were sacrificed by CO_2_ asphyxiation and subcutaneous tissues’ (approximately 4 cm^2^) nearby injection sites were carefully excised. Each tissue was fixed with a pin and immersed into 10% neutral-buffered formalin for 72 h. The immobile tissues were rinsed with 70, 80, 95, and 100% alcohol and cleared with xylene. The dehydrated samples were embedded in paraffin wax at 40 °C, with a shaking speed of 100 rpm. The specimens embedded in paraffin blocks were then sliced using a microtome (Model Leica RM2165, Wetzlar, Germany) at a thickness of 20 μm. The sections on the glass slide were washed twice with xylene and subsequently dehydrated with ethanol treatment. The sections were stained with hematoxylin and eosin (H&E) for 10 min. The H&E-stained samples mounted onto slides covered with a mounting medium (Canadian balsam mounting solution) were then observed using a Pannoramic 250 Flash digital microscope (P250 Flash digital microscope; 3DHISTECH, Budapest, Hungary) equipped with CaseViewer software (3DHISTECH, Budapest, Hungary). Histopathological evaluation of the injected subcutaneous tissues included the shape and size the depot, angiogenesis, fibrosis, necrosis, and the degree of infiltration of inflammatory cells such as macrophages and lymphocytes.

### 2.6. In Vivo Pharmacokinetic Profile of RTG Crystalline Suspensions in Rats

The drug concentration-time profile of RTG following SC injection of AS or OS was determined in normal rats after the approval of the Institutional Animal Care and Use Committee (IACUC) of Dankook University (Cheonan, Korea) (DKU-19-032, 8 October 2019). Six-week-old male Sprague-Dawley rats (200 ± 20 g) acquired from Samtako Bio Korea (Gyeonggi-do, Korea) were kept under constant temperature (23 ± 1 °C) and light cycle (day/night: 12 h). After 3 d of acclimatization, RTG-loaded AS or OS was subcutaneously administered to the back scruff of the normal rats (30 mg/kg as RTG) through a 26-gauge needle-equipped syringe (*n* = 5 per group). At predetermined times, blood samples (approximately 600 µL) were withdrawn from the jugular vein using a 26-gauge needle-equipped syringe pretreated with heparin (20 IU/mL). The collected blood was centrifuged at 13,000 rpm for 10 min to obtain plasma. The collected plasma samples were then kept in a deep freezer at −80 °C until LC-MS/MS analysis.

To determine the concentration of RTG in plasma, the plasma was treated and analyzed using Kim et al.’s method with some modification [33]. Briefly, an aliquot (20 μL) of the plasma was mixed with 500 μL of methyl tert-butyl ether. Then, the whole top layer obtained from the centrifugation of the mixture was dried at room temperature and 5 μL of the solution reconstituted in 100 μL of acetonitrile was applied to LC-MS/MS analysis. A matrix-matched standard (MMS) and a standard-spiked sample (SSS) were prepared by spiking an appropriate volume of a RTG standard solution into the final plasma extract obtained from a blank plasma and into a blank plasma prior to the sample treatment, respectively. For LC, a Shimadzu Nexera UPLC system (Tokyo, Japan) with a Waters Atlantis HILIC Silica column (2.1 × 150 mm, 3 µm, Milford, MA, USA) was employed. The mobile phase was the mixture of 2 mmol/L of an aqueous ammonium formate solution (0.1% *v*/*v* formic acid) and acetonitrile (0.1% formic acid) at a volume ratio of 15:85. The column oven temperature and flow rate were set at 40 °C and 0.25 mL/min, respectively, with a running time of eight minutes. The column eluent was delivered to a Shimadzu LCMS 8060 triple quadrupole mass spectrometer through electrospray ionization in positive ion mode. Mass analysis was carried out through multiple reaction monitoring (MRM) and MRM transitions of RTG are as follows: 316.1 *m*/*z* (precursor ion)/147.3 *m*/*z* (product ion)/−24 V (collision energy), 316.1 *m*/*z*/77.1 *m*/*z*/−73 V, and 316.1 *m*/*z*/107.1 *m*/*z*/−64 V were the screening transition (for quantitation), the confirmatory transition 1, and the confirmatory transition 2, respectively. To quantitate RTG in plasma, areas of the screening transition peaks from sample analyses were compared to calibration curves built using those from MMS analyses, and a couple of preconditions were checked prior to quantitation. First, all three transition peaks of RTG needed to have the same retention time (the identity confirmation). Additionally, the signal-to-noise ratio values of the screening transition peak and the confirmatory transition peaks needed to be at least 10 and at least 3, respectively (the sensitivity test). All peaks mentioned over the manuscript satisfied the identity confirmation and the sensitivity test. A partial validation of the present method was successfully carried out in the aspects of specificity, linearity, accuracy, precision, and sensitivity. First, specificity was confirmed by the absence of the RTG screening transition peak at the retention time of RTG from a blank matrix (negative control) result. Additionally, all calibration curves built by using MMSs (0.1, 2, 10, 25, 50, and 100 ng/mL, *n* = 3) showed good linearity (the coefficient of determination values of 0.996 ± 0.0004). Third, recovery (the division of the screening transition peak area of a SSS by that of its counter MMS) values evaluated at 0.3, 40, and 80 ppb were 96.60 ± 1.78, 99.69 ± 1.36, and 98.27 ± 3.17, respectively, and they are good enough to support its good accuracy and precision. Finally, the lower limit of quantitation (LLOQ), a parameter of sensitivity, was determined to be 0.3 ng/mL, the lowest concentration showing good R values within the linear dynamic range. Pharmacokinetic parameters, such as the area under the curve for drug concentration in plasma–time from zero to 21 days (AUC_0–21 days_), maximum drug concentration in plasma (C_max_), time to reach C_max_ (T_max_), and elimination half-life (T_1/2_), were calculated using a WinNonlin^®^ version 5.2 program (Pharsight Co., Mountain View, CA, USA).

### 2.7. Statistical Analysis

All experiments were duplicated at least thrice, and the data are presented as mean ± standard deviation (SD). The differences in pharmacokinetic parameters between RTG OS and AS were statistically analyzed using Student’s *t*-test. A *p*-value less than 0.05 signified statistical significance.

## 3. Results and Discussion

### 3.1. Selection of Dispersant and Control of Particle Size of RTG AS System

We aimed to design RTG-loaded AS and OS and compare their pharmacokinetic patterns, along with the in vivo behavior of drug crystals at the injection site. The RTG-loaded AS and OS systems were fabricated using a bead-milling technique, a wet milling approach, decreasing drug particle size to sub-micrometer in aqueous and oily vehicles. Compared to other nanosizing techniques, the mechanical grinding technique provides several advantages such as low energy utilization, ease of scale-up, low batch-to-batch variation, and no use of organic solvents [34,35,36]. In the present study, dual centrifugation, a laboratory-scale wet ball milling technology, was employed to pulverize the RTG raw material using the additional rotation of the samples during the centrifugal process with zirconia beads. The strong collision of the drug particles with the beads inside the vials provides rapid size reduction of the drug particles in the vehicles. Hagedorn et al. (2019) [37] reported that the laboratory-scale dual centrifugation method provided comparable particle reduction and homogeneity of drug nanosuspensions to those obtained through larger-scale agitator mills. In designing the RTG crystalline suspension, the drug content in the formulation was set to 100 mg/mL. When the crystalline suspension was fabricated over 150 mg/mL, the suspension was drastically thickened, causing difficulties in SC injection via a 26-gauge syringe. Nevertheless, the high-payload crystalline system can afford a bolus dose equivalent to over 2 weeks of transdermal RTG therapy (2–8 mg per day), respecting that up to 1.5 mL of injection volume is allowed for SC administration.

First, numerous hydrophilic polymers and surfactants were screened as suspending agent to formulate the AS system of lipophilic compounds (log*P* = 4.7) [38]. The concentration of the dispersant, milling intensity, and milling time was fixed at 5 mg/mL, 1500 rpm, and 1.5 h, respectively. The ASs formulated with Poloxamer 188, kolliphor HS 15, PEG 4000, and PVP K17 were inconsistent and highly viscous, making them unsuitable for injection (Table 1). In contrast, ASs prepared with Na. CMC, Tween 80, kolliphor RH 40, and kolliphor EL were uniform, with particle sizes < 500 nm. The mean diameters of the crystalline suspensions were 374.8, 414.7, 447.5, and 424.4 nm, respectively, with low PDI values of 0.320 (Table 1). In particular, Na. CMC, a cellulose derivative, provided the finest particle size with appropriate homogeneity (PDI value of 0.206), indicating that the cellulose derivative included in the colloidal system was attached to the drug particular surface and consequently impeded crystal aggregation via steric hindrance. The hydrogen bonding interaction of the carboxylic group with the amino group of the solid-state drug particles, a hydrogen bonding donor, might encourage the adsorption of the cellulose derivative onto the surface of the RTG nanoparticles. Therefore, Na. CMC was chosen as the suspending agent for further RTG-AS preparation.

The crystal size of the RTG-loaded AS was adjusted by controlling the concentration of the dispersant and the bead-milling speed (Figure 1A). The concentration of dispersant and milling intensity have a great influence on particle size and uniformity of crystalline suspension [19,20]. As expected, the concentration of dispersant (Na. CMC) increased and the particle size decreased at the same milling intensity. When the milling strength was set to 500 rpm, the particle size was reduced to 1747, 854, and 670 nm with Na. CMC concentrations of 0.5, 1.0, and 3.0 mg/mL, respectively. However, the effect of milling force on particle size tended to depend on the amount of dispersant. When the concentration of dispersant was 0.5 mg/mL, the particle size increased from 1.7 µm to 4.5 µm as the milling intensity increased from 500 rpm to 1500 rpm. The dispersant could not sufficiently cover the surface of the individual pulverized drug particles in an environment in which the dispersant was insufficient (0.5 mg/mL), causing agglomeration of the particles. In contrast, when the concentration of Na. CMC was as much as 3.0 mg/mL, the particle size decreased from 670 to 407 nm as the milling intensity increased, effectively surrounding the crushed drug nanoparticles with the anionic hydrophilic polymer. To evaluate the effect of the dispersion medium, the particle size of the AS system was adjusted to 800–900 nm, similar to that of the OS system, which was fabricated with 0.1% *w*/*v* Na. CMC at a milling speed of 500 rpm.

### 3.2. Selection of Oily Vehicle and Control of Particle Size of RTG-Loaded OS System

Injectable oily vehicles were screened to formulate an RTG OS system based on particle size, homogeneity, and drug solubility (Table 2). In formulating this preparation, the low solubility of the lipophilic compound (log*P* value of 4.7) in oily vehicles was desired, as it reduced the initial burst release. OS systems prepared with medium-chain triglycerides (Miglyol 810N and tricaprylin) were uniform, with appropriate flow properties. However, because of the high solubility in these medium-chain triglycerides when the OS was designed at RTG concentrations of 100 mg/mL, 44.0 and 48.6% of the dose remained in a dissolved state, respectively. The OS systems prepared with cottonseed oil, soybean oil, corn oil, sesame oil, and peanut oil were also homogeneous, with particle sizes ranging from 831 to 945 nm. The fractions of RTG dissolved in these oily vehicles were 21.2, 22.3, 22.9, 19.6, and 20.8%, respectively, with appropriate flowability, providing easy filling and injection through a 26-gauge needle-equipped syringe; because of its homogeneity and because it has the lowest drug solubility, sesame oil was employed as the oily vehicle for the preparation of OS.

The effects of milling speed and time on crystal size in the RTG OS were further evaluated (Figure 1B). As expected, as the milling intensity and time increased, the crystal size in the sesame oil decreased; when milling speed was increased to 1000, 1500, and 2000 rpm with 1.5 h of milling time, the RTG particle size progressively decreased to 6.3 and 1.2 μm. In addition, as the milling time increased, the particle size declined; when milling at 2000 rpm as the milling time was expanded from 0.5 h to 1.5 h, the particle size decreased from 1275 nm to 878 nm. A nanosized OS system (800–900 nm) fabricated with milling speed and time of 2000 rpm and 1.5 h, respectively, was employed for further in vivo pharmacokinetic evaluation with the AS system.

### 3.3. Morphological and Physical Characteristics of RTG-Loaded AS and OS

We designed two RTG crystalline suspensions (AS and OS) with comparable particle diameters, and characteristics of injectable suspensions were evaluated by aspect of morphology, drug content, crystal size, viscosity, and crystallinity (Table 3 and Figure 2). The particle sizes were adjusted to 853 nm for AS and 878 nm for OS, respectively, which is within the size range of marked crystalline injectable suspensions. The mean particle sizes of paliperidone palmitate (Invega sustenna^®^, Janssen, NJ, US), aripiprazole (Abilify maintena^®^, Otsuka Pharmaceutical Inc., Tokyo, Japan), rilpivirine suspensions (Cabenuva^®^, ViiV Healthcare, GL, Research Triangle, NC, UK), and olanzapine (Zyprexa Relprevv^®^, Eli Lilly, Indianapolis, IN, USA) was informed to be <1000 nm, 1–10 µm, <500 nm, and <3 µm, respectively [39,40,41,42]. The morphological features of the RTG nanocrystals suspended in aqueous or oily vehicles were examined by Cryo-TEM (Figure 2A,B). In both formulations, the crystals generally exhibited a rectangular or hexagonal crystalline form. Although the vehicles were different, the identical pulverization process provided a comparable crystalline shape. RTG concentrations in AS and OS were 100.9 mg/mL and 100.7 mg/mL, respectively, indicating no degradation of RTG during fabrication (Table 3). The amount of RTG dissolved in the OS was 19.6 mg/mL, which is markedly higher than AS (0.07 mg/mL), but over 80% of the initial load was suspended as solid state in the oily vehicle (Table 3).

Powder XRD analysis showed that the RTG nanocrystals suspended in AS and OS possess distinctive peaks at 11.8°, 13.4°, 16.9°, 17.5°, 18.7°, 20.3°, 21.7°, and 22.7°, identical to that of the raw material prior to milling (Figure 2C). This indicates that in both types of vehicles, the nanocrystals have identical crystalline lattices, preserving the intrinsic crystallinity of the raw material during the nanomilling process. The apparent viscosity curves of the RTG-loaded AS and OS, determined using a rotational rheometer, are depicted in Figure 2D. Both types of vehicles are Newtonian fluids, and thus exhibit constant viscosity regardless of the shear rate. In contrast, the RTG AS and OS systems containing drug nanoparticles at a high concentration (100 mg/mL) exhibited pseudoplastic patterns, and their viscosity decreased as the shear rate increased. The apparent viscosities of AS and OS were 73.8 and 338.4 centipoise (cP) at shear rates of 10 s^−1^, and 6.9 and 150.9 cP at 100 s^−1^, respectively (Table 3 and Figure 2D). Although the OS system prepared with sesame oil was more viscous than AS, both formulations could be filled and administered without difficulty using 26-gauge syringes.

### 3.4. In Vitro Dissolution Profile of RTG-Loaded AS and OS

The in vitro dissolution patterns of RTG from the different vehicles were comparatively evaluated using the dialysis bag method (molecular weight cut-off: 50 kDa). To provide sink conditions for drug release and the miscibility of sesame oil with dissolution media, a mixture of phosphate-buffered saline (pH 7.4) and 2-propanol (6:4 *v*/*v*) was employed [43,44]. The solubility of the drug in phosphate-buffered saline was determined to be 1 mg/mL, whereas that in the dissolution medium was 100 mg/mL. Thus, 200 mL of dissolution media provided sufficient solubility, warranting sink conditions for dissolution tests of the water-insoluble compound.

Under sink conditions, over 50% of the drug loaded into the AS system was released after 24 h, followed by over 90% drug release after 48 h (Figure 3). The dissolution of RTG from OS was slightly slower than that of AS; approximately 42% of the drug was released after 24 h, and it took 96 h for drug release to exceed 90%. Despite the larger amount of drug dissolved in the oily vehicle, the dissolution was delayed, probably due to the additional partitioning process of the drug particles and/or molecules from the oily vehicle into aqueous media and partial obstruction in the passage of drug molecules through the membrane pores with the oily vehicle. The two formulations showed analogous dissolution patterns; the $f_{2}$ value between AS and OS was calculated to be 51.1, indicating that the dissolution pattern of the two formulations was comparable with identical drug crystal size and crystallinity.

### 3.5. Macroscopic and Histopathological Observation of Injection Site following SC Injection

The extent and rate of drug absorption following injection of drug crystalline suspensions have been reported to be markedly affected not only by their own physicochemical properties, such as drug solubility and dissolution rate, but also by the in vivo performance of drug particles, such as drug aggregation and inflammatory responses at the injection site [8,45]. However, most studies have been limited to AS and, to date, there have been few that compared local behavior in relation to the dispersion medium, as well as the pharmacokinetic profile. Therefore, the differences in drug behavior at the injection site were comparatively evaluated to further interpret the pharmacokinetic behavior of RTG OS and AS.

Macroscopic observations revealed a noticeable difference in the degree of drug aggregation and local inflammation between AS and OS (Figure 4A). Upon SC administration of AS, agglomeration of drug particle and the shape of closely packed clusters of particles in the tissue were observed. The increase in surface free energy due to the rapid absorption of the aqueous suspension vehicle might cause drug aggregation at the injection site. Conversely, no distinct drug aggregates were observed with OS, with the Sudan III-stained delivery vehicle spreading widely at the injection site (Figure 4A). Oily vehicles might be adsorbed onto the surface of the RTG particles, maintaining their suspended state at the injection site. Seven days post-dosing, white drug aggregates ranging from 5 to 10 mm and surrounding inflammatory cells and fibrous matrix were observed at the AS-injected site. In contrast, no marked drug aggregates were distinctively observed on the seventh day after OS injection, with the red-colored oily vehicle remaining at the injection site (Figure 4A). The prolonged retention of sesame oil, an oily vehicle, was consistent with previous findings that the elimination T_1/2_ of castor oil, fractionated coconut oil, isopropyl myristate, and sesame oil was approximately 20, 14, 20, and 23 days, respectively, after SC or IM injection in pigs [46]. Moreover, an oil depot composed of sesame oil and benzyl alcohol (9:1 *v*/*v*) was visible on histopathological evaluation, even 4 weeks after IM injection [47].

These differences in drug distribution and biological response to RTG nanocrystals at the injection site were also examined histopathologically (Figure 4B). When normal saline was administered, no irritation or damage was observed at the injection site 2, 7, and 21 days post-dosing. However, when the RTG AS system was administered, a marked foreign-body reaction was observed at the injection site. On the second day of injection, inflammatory bands chiefly comprised of polymorphonuclear leukocytes (PMNs) and lymphocytes were observed, granulating the drug aggregates. Seven days after dosing, the inflammatory area had gradually decreased, with a distinct infiltration of chronic inflammatory cells, including lymphocytes and macrophages. After activation of the macrophages upon adherence to the drug particle, fibrosis (i.e., fibrous encapsulation) enclosed the drug crystals with interfacial foreign-body reactions. Mild-to-moderate angiogenesis was also observed on day 7. The time course of the local responses against RTG AS was consistent with a previous report [48] that the local response to intramuscularly injected paliperidone palmitate nanoparticles was principally acute inflammation, granulomatous inflammatory reaction with extensive infiltration of inflammatory cells, fibrosis, and local angiogenesis. Three weeks after AS injection, most local inflammatory reactions and necrosis at the subcutaneous area were markedly alleviated, as RTG crystals were dissolved and absorbed into the bloodstream. Unfortunately, the OS injection site vanished during the H&E staining process, so the injection site could not be closely observed. The sesame oil remaining in the tissue might have faded away during pretreatment with xylene, ethanol, etc. Nevertheless, from the H&E images with the OS system, we could deduce that a considerable amount of sesame oil might remain at the injection site, with no formation of hard or thickened depots such as the drug aggregates or fibroblastic band, as previously shown in the macroscopic image (Figure 4A). The vanished area in H&E images 21 days post-dosing of OS was markedly reduced, as the oily vehicle might have been gradually removed from the injection site.

### 3.6. In Vivo Pharmacokinetic Profile of RTG following SC Injection of AS or OS in Rats

The drug concentration-time profile in plasma following the SC administration of AS and OS (20 mg/kg as RTG) is depicted in Figure 5. The corresponding pharmacokinetic parameters, including C_max_, T_max_, AUC, and elimination T_1/2_, are presented in Table 4. In current practice, RTG patches are prescribed at 2–8 mg/60 kg/day for early stage Parkinson’s patients, and up to 16 mg/60 kg/day for patients with advanced-stage disease [49]. In clinical trials, approximately 35% of the administered dose was transdermally absorbed with a trough concentration (C_trough_) of 0.1 ng/mL [49]. Therefore, in our study, RTG was SC administered at a dose of 20 mg/kg, expecting to maintain a C_trough_ or higher RTG concentration over 2 weeks, following a single administration.

After SC injection of both AS and OS, the level of RTG in plasma increased rapidly (Figure 5), reaching C_max_ (T_max_) at 0.28 and 0.43 days, respectively (Table 4). Drug dissolution may proceed promptly and profoundly, with acute edema and swelling at the injection site following RTG injection [45]. Although T_max_ was comparable between AS and OS, there was a marked difference in C_max_ values; C_max_ obtained with the OS system (103.43 ng/mL) was significantly higher (about 1.7-fold) than that of AS (61.5 ng/mL) (Table 4). After reaching C_max_, the plasma concentration of RTG following both AS and OS injection decreased sharply for 2 days, with elimination T_1/2_ of 0.76 and 0.54 days, respectively (Figure 5 and Table 4). After RTG OS injection, the drug concentration in the plasma continued to rapidly decrease, and the drug concentration was determined to be below the LOQ 4 days post-dosing (Figure 5). In contrast, the plasma concentration of RTG gradually decreased following AS injection, maintaining over 0.5 ng/mL in the plasma, even after 21 days, and falling below the LOQ after 4 weeks. Despite the noticeably different drug concentration-time profiles between AS and OS, there was no significant difference in the AUC value, a parameter representing total drug exposure level, between AS (95.1 ng·day/mL) and OS (104.3 ng·day/mL) (Table 4). These pharmacokinetic results indicate that the two formulations with different delivery vehicles exhibited an equivalent degree of drug absorption, with marked differences in the rates of drug absorption and elimination.

This marked difference in the absorption rate was highly correlated with the local behavior of the RTG particles observed previously. As shown in Figure 4A, the oily vehicle stained with Sudan III (red color) showed extended retention at the injection site, compared to Trypan-blue-strained AS. Thus, the higher solubility of RTG in sesame oil compared to the aqueous vehicle caused a greater difference in the dissolution rate at the in vivo injection site than in the in vitro dissolution test, promoting the dissolution of the drug crystals at the injection site with extended retention of the oily vehicle. This difference in the dissolution rate at the injection site in vivo markedly affected the absorption rate of the drug, resulting in different pharmacokinetic profiles in rats.

Interestingly, OS did not cause drug aggregation at the injection site, resulting in faster dissolution and absorption into the bloodstream. The oily vehicle attached to a particular surface might aid in maintaining the uniform distribution of the crystals at the injection site. Conversely, in the case of the AS system, the aqueous vehicle might be rapidly absorbed immediately after administration, rapidly escalating the surface free energy of the particles, thus forming drug aggregates over 5 mm in diameter at the injection site. The formation of agglomerates markedly retarded drug dissolution at the injection site, in accordance with the Noyes–Whitney equation: dM/dt = k·S·Cs, where dM/dt is the dissolution rate, k is the rate constant, S is the surface area of the drug particles, and Cs is drug solubility [50,51]. The increase in drug particle size led to a marked reduction in surface area, thereby slowing RTG dissolution at the injection site.

Moreover, the formation of drug clusters following AS injection was subsequently accompanied by a foreign body reaction (FBR) at the injection site. As shown in Figure 4B, the drug cluster was covered with a thick macrophage-rich fibrous matrix two days after dosing. Tissue damage and extravasation of blood caused by RTG AS injection might have triggered an immediate rush of inflammatory-mediating cells to the injection site. Because of the large size of the RTG aggregates, macrophages and proteins, such as albumin and fibrinogen, derived from extravasated blood, were unable to phagocytose the entire implant and became non-specifically adsorbed to the surface of the drug aggregates, forming a macrophage layer around them. It has been reported that IM injection of paliperidone palmitate nanocrystals suspended in an aqueous vehicle caused foreign body granulomas, displaying a central, almost cell-free depot, which was surrounded by a dense inflammatory rim at 48 h after injection [45]. The capsule mainly consisted of epithelioid macrophages and included some T cells, young fibroblasts, vascular endothelial cells, and a few granulocytes [45]. Subsequently, the chronic stage of FBR instigated the encapsulation of RTG aggregates in a layer of fibrous tissue. This fibrotic layer acts as a barrier to contact between drug aggregates and interstitial fluid and/or blood, delaying the dissolution of RTG and its liberation into the bloodstream and providing a protracted pharmacokinetic profile following SC injection of AS.

In the present study, we revealed the importance of observing the local behavior of drug particles and the impact of vehicles when designing drug crystalline suspension-based long-acting delivery systems. In designing a long-acting delivery system, an aqueous vehicle-based drug suspension was more advantageous than an oily vehicle-based system with a protracted pharmacokinetic profile. Nevertheless, it is essential to evaluate the degree of local inflammation and pharmacokinetic profile after multiple administration, as Parkinson’s disease is a chronic disease. Moreover, as the non-ergot dopamine agonist has a narrow safety margin [52], more delicate drug concentration profile is required, along with the assessment of local tolerability. We are now designing drug crystal-incorporated aqueous hydrogel systems to precisely control drug concentration profiles over 4 weeks following a single injection. After establishing the designed formulation, we are planning to evaluate the pharmacokinetic profile and local inflammatory response after repeated administration. Next, the complex and multiphasic in vivo drug dissolution pattern of the RTG LAI at injection site will be simulated with pharmacokinetic profile.

## 4. Conclusions

In this study, aqueous and oily vehicle-based LAI suspensions of RTG were designed, and their pharmacokinetic profiles were comparatively evaluated. We found that the delivery vehicle had a marked effect on local drug solubility, drug particle aggregation, and biological response against the drug crystals at the injection site. The aqueous vehicle-based RTG crystalline suspension provided protracted pharmacokinetics compared to the OS system, with lower solubility at the injection site, formation of drug aggregates, and even granulation with fibroblastic bands. Most necrosis and inflammation responses at the injection site recovered by 3 weeks after AS injection. We are currently investigating an AS-based advanced LAI system to provide a steady pharmacokinetic profile over 4 weeks following a single injection.

## Figures and Tables

**Figure 1 pharmaceutics-14-02630-f001:**
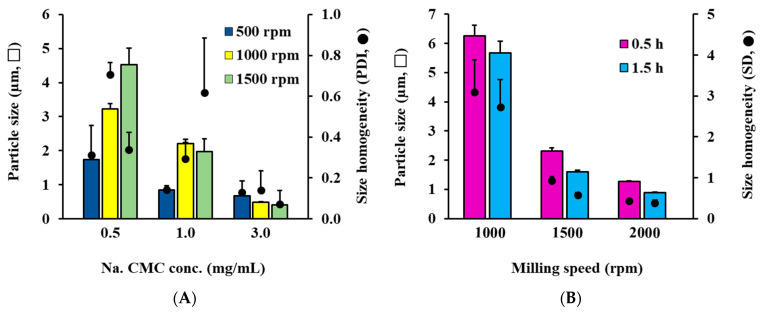
Effect of formulation and fabrication parameters on drug particle size and homogeneity in RTG-loaded injectable crystalline suspension. (**A**) Effect of Na. CMC concentration and bead-milling speed on drug particle size and homogeneity of the AS system. (**B**) Effect of bead-milling speed and milling time on crystal size and homogeneity of the OS system. Notes: The particle size and homogeneity of AS system were determined using dynamic light scattering method. On the other hand, for the RTG-OS, the size of the suspended RTG crystals was determined using a micrometer for 200 randomly selected RTG particles from light microscope images, with difficulty in analyzing hydrodynamic diameter with dynamic light scattering method (Zetasizer Nano^®^).

**Figure 2 pharmaceutics-14-02630-f002:**
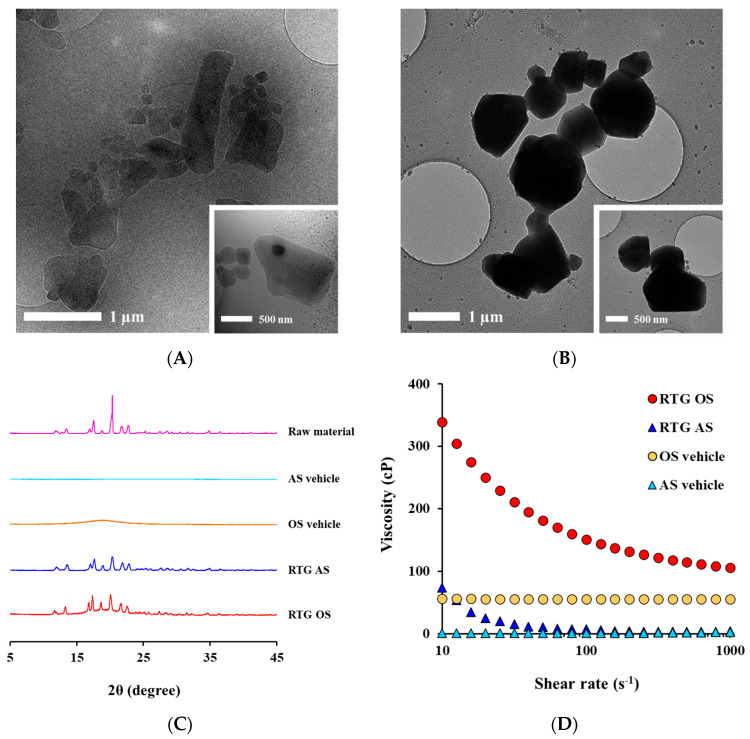
Morphological and physical features of RTG-loaded AS and OS systems. Transmission electron microscope (TEM) images of drug particles in (**A**) AS and (**B**) OS. (**C**) X-ray diffraction (XRD) patterns of the raw material, delivery vehicles, and injectable drug suspensions. (**D**) Apparent viscosity curve of RTG-loaded drug suspensions and delivery vehicles at a shear rate of 10–1000 s^−1^.

**Figure 3 pharmaceutics-14-02630-f003:**
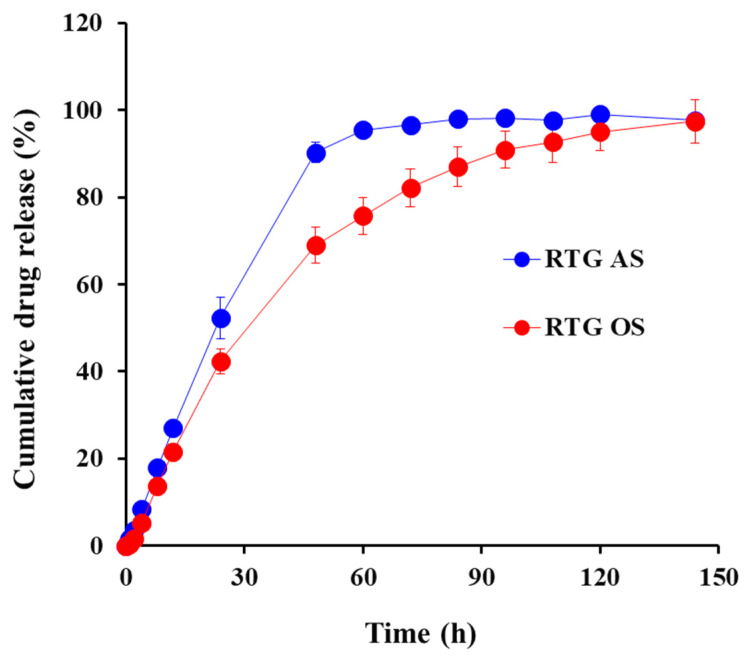
In vitro dissolution profile of RTG from AS (🔵) and OS (🔴) embedded in dialysis tube (molecular weight cut-off: 50 kDa) under sink conditions. Notes: The sink conditions were provided by adding 2-propanol to phosphate-buffered saline (pH 7.4) (40:60 *v*/*v*).

**Figure 4 pharmaceutics-14-02630-f004:**
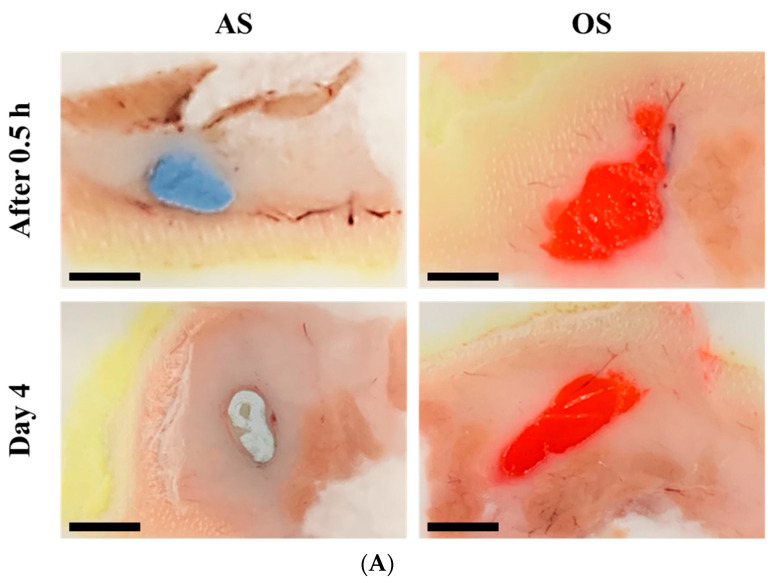
Representative macroscopic and histological images of rat back scruff tissues at 2, 7, and 21 days after SC injection of RTG AS or OS systems (30 mg/kg as RTG). (**A**) Macroscopic images of injected sites at 2, 7, and 21 days after SC injection of RTG AS or OS. (**B**) Histological observation of the HE-stained back scruff tissues followed by SC injection of negative control (normal saline) and RTG OS and AS systems. Notes: (**B**) The depots formed by the injection of RTG AS are indicated by (*) and the injection sites of OS are indicated by red arrows. Black arrows indicate granulocyte or macrophage infiltration (i) and active capillaries or angiogenesis (a). Scale bars in (**A**) and (**B**) represent 5 mm and 1 mm, respectively.

**Figure 5 pharmaceutics-14-02630-f005:**
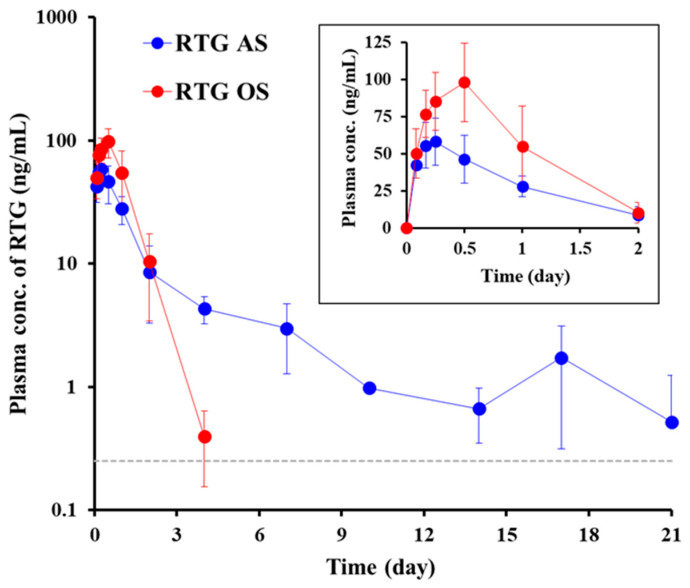
Plasma concentration-time profiles of RTG following SC injection of RTG AS (🔴) and RTG OS (🔵) in rats (30 mg/kg as RTG). Notes: Data are expressed as mean ± SD (*n* = 5). The dashed line (--) indicates the limit of quantitation (LOQ). The plasma concentration of RTG after AS or OS injection was below the LOQ after 28 and 7 days, respectively.

**Table 1 pharmaceutics-14-02630-t001:** Assessment of suspending agent on dispersibility, particle size, and homogeneity of RTG crystals in aqueous vehicle.

Stabilizer ^a^	Appearance ^b^	Particle size (nm) ^c,d^	Homogeneity (PDI) ^d,e^
Na. CMC	Homogeneous	374.8 ± 1.68	0.206 ± 0.015
Tween 80	Homogeneous	414.7 ± 3.39	0.319 ± 0.017
Poloxamer 188	Homogeneous	1099 ± 303.6	0.334 ± 0.279
Kolliphor HS 15	Homogeneous	3021 ± 222	0.407 ± 0.035
Kolliphor RH 40	Homogeneous	447.5 ± 5.69	0.236 ± 0.082
Kolliphor EL	Homogeneous	424.4 ± 4.49	0.295 ± 0.027
PEG 4000	Aggregated	7148 ± 715.5	0.564 ± 0.121
PVP K17	Aggregated	3198 ± 130.8	0.665 ± 0.197

^a^ The concentrations of RTG powder and stabilizer in the aqueous vehicle (15 mM phosphate buffer, 0.9% *w*/*v* of normal saline, pH 7.0) were fixed to 100 and 5 mg/mL, respectively. The milling intensity and time was set to 1500 rpm and 1.5 h, respectively. ^b^ Visually evaluated. ^c^ Mean hydrodynamic size measured using dynamic light scattering method (Zetasizer Nano^®^ Instruments, Malvern, UK). ^d^ Data represent mean ± SD (*n* = 3). ^e^ Polydispersity index, calculated by following equation: $PDI={\left(\sigma /d\right)}^{2}$, where σ and d indicate the standard deviation and mean particle diameter, respectively.

**Table 2 pharmaceutics-14-02630-t002:** Dispersibility, particle size, and solubilized fraction (%) of RTG OSs prepared with different oily vehicles using bead-milling technology.

Vehicle ^a^	Appearance ^b^	Particle size (nm) ^c,d^	Fraction Dissolved (% *w*/*w*) ^d^
Castor oil	Homogeneous	945.3 ± 19.2	27.2 ± 2.8
Cottonseed oil	Homogeneous	831.0 ± 21.2	21.2 ± 0.2
Soybean oil	Homogeneous	896.5 ± 44.5	22.3 ± 0.2
Corn oil	Homogeneous	835.4 ± 38.3	22.9 ± 0.0
Sesame oil	Homogeneous	878.2 ± 28.1	19.6 ± 0.0
Peanut oil	Homogeneous	835.7 ± 16.1	20.8 ± 0.1
Miglyol 810N	Homogeneous	724.2 ± 9.7	45.9 ± 1.0
Tricaprylin	Homogeneous	3234.5 ± 576.4	48.6 ± 0.1

^a^ The concentration of RTG in the oily vehicle was set to 100 mg/mL with a bead-milling speed of 2000 rpm for 1.5 h. ^b^ Visually evaluated. ^c^ Determined using a micrometer for 200 randomly selected RTG particles from light microscope images. ^d^ Data are expressed as mean ± SD (*n* = 3).

**Table 3 pharmaceutics-14-02630-t003:** Physicochemical characteristics of RTG-loaded AS and OS formulations.

Parameters	AS	OS
RTG conc. (mg/mL) ^a^	100.9 ± 0.09	100.7 ± 0.53
Suspended (mg/mL) ^a^	100.8 ± 0.09	81.1 ± 0.53
Dissolved (mg/mL) ^a^	0.07 ± 0.00	19.6 ± 0.04
Particle size (nm) ^a^	853.7 ± 19.9 ^b^	878.3 ± 28.1 ^c^
Zeta potential (mV) ^d,a^	−39.97 ± 1.46	not determined
Viscosity (cP) ^e^	73.8	338.4

^a^ Data are expressed as mean ± SD (*n* = 3). ^b^ Mean hydrodynamic size determined using dynamic light scattering (Zetasizer Nano^®^ Instruments). ^c^ Determined using a micrometer for randomly selected 100 RTG particles from the light microscope images. ^d^ Determined using Zetasizer Nano^®^ Instrument. ^e^ Determined at 25 °C and a shear rate of 10 s^−1^.

**Table 4 pharmaceutics-14-02630-t004:** Pharmacokinetic parameters of RTG following SC injection of drug suspensions in rats (30 mg/kg as RTG).

Parameters	RTG AS	RTG OS
AUC_(0–21 days)_ (ng·day/mL)	95.1 ± 6.4	104.3 ± 28.0
C_max_ (ng/mL)	61.5 ± 12.1	103.4 ± 22.7 *
T_max_ (day)	0.28 ± 0.13	0.43 ± 0.15
T_1/2α_ ^a^ (day)	0.76 ± 0.28	0.54 ± 0.18
T_1/2β_ ^b^ (day)	2.95 ± 0.66	Not determined

^a^ Time required for the level of RTG in plasma to decline to half of the C_max_. ^b^ Time required for half of the RTG in plasma to be removed during the terminal elimination phase. The elimination phase lasted from 2 to 21 days. Notes: Data are expressed as mean ± SD (*n* = 5). Significantly different from RTG AS (* *p* < 0.05) by Student’s *t*-test.

## Data Availability

Not applicable.

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
