# Peer review of "Effect of Dispersion Medium on Pharmacokinetic Profile of Rotigotine Crystalline Suspension following Subcutaneous Injection"

_pharmaceutics, 2022, doi:10.3390/pharmaceutics14122630_

Round 1

Reviewer 1 Report

The authors have presented a well-conceived and well-designed study and analysis which present a comprehensive approach to delivering better performing formulation for Parkinsons patients.

This review has a few minor comments for the authors' consideration:

1. On Table 1, the homogeneity factor could be better described. In the footnotes this is described as polydispersity index and its not clear how the values are arrived at using the calculation method described. Suggest to describe the exact equation used to arrive at the values in the last column under "homogeneity".

2. Figure 1 presents different quantities for the 2 formulations making it difficult to compare. For AS, the effect of dispersant is analyzed whereas for OS the impact of milling speed. Some clarity is required on why this is done in this way. Also the homogeneity presented is different on Figure 1a (PDI) and on Figure 1b(SD). 

3. The authors have observed an acute local inflammatory response at the site of injection with the AS formulation which seemed to resolve in 21 days. However, if approved, Parkinsons patients would be taking this patch in a chronic fashion. Suggest to consider a multiple dosing study in rats to look at the PK of the drug after 2-3 doses of the AS and how the drug concentrations are maintained. Is there a safety limit below which plasma concentrations need to be which could be surpassed after multiple doses? Also with multiple application of patches this acute inflammation might worsen. Could the authors discuss this based on literature evidence and include this in discussion as future work?

Reviewer 2 Report

The article is novel in the sense that it presents alternative solutions to the problem of administering Rotigotine in topical patches through formulations in oily and aqueous suspensions.

This type of work is always interesting, given that alternatives to public health problems are sought.

The experimental protocols shown are adequate for the proposed objective. I see no problems in the characterization of the particles of both formulations. The in vivo experiments have the approval of a bioethics committee, and the appropriate quantification methodology. It would be very appropriate to show if the analytical methods used by mass-HPLC were validated. Pharmacokinetic studies were adequately developed.

The release profiles show a relatively similar behavior, assessed with the factor F2. However, the pharmacokinetic results show a marked difference with respect to the formulations. All of this is mentioned in their discussion and is technically supported by experimentation and analytical methods.
